# Learning Linear Dynamical Systems
# via Spectral Filtering

**Elad Hazan, Karan Singh, Cyril Zhang**
Department of Computer Science
Princeton University
Princeton, NJ 08544
{ehazan,karans,cyril.zhang}@cs.princeton.edu

## Abstract

We present an efficient and practical algorithm for the online prediction of discrete-time linear dynamical systems with a symmetric transition matrix. We circumvent the non-convex optimization problem using improper learning: carefully overparameterize the class of LDSs by a polylogarithmic factor, in exchange for convexity of the loss functions. From this arises a polynomial-time algorithm with a near-optimal regret guarantee, with an analogous sample complexity bound for agnostic learning. Our algorithm is based on a novel filtering technique, which may be of independent interest: we convolve the time series with the eigenvectors of a certain Hankel matrix.

## 1 Introduction

Linear dynamical systems (LDSs) are a class of state space models which accurately model many phenomena in nature and engineering, and are applied ubiquitously in time-series analysis, robotics, econometrics, medicine, and meteorology. In this model, the time evolution of a system is explained by a linear map on a finite-dimensional hidden state, subject to disturbances from input and noise. Recent interest has focused on the effectiveness of recurrent neural networks (RNNs), a nonlinear variant of this idea, for modeling sequences such as audio signals and natural language.

Central to this field of study is the problem of *system identification*: given some sample trajectories, output the parameters for an LDS which generalize to predict unseen future data. Viewed directly, this is a non-convex optimization problem, for which efficient algorithms with theoretical guarantees are very difficult to obtain. A standard heuristic for this problem is expectation-maximization (EM), which can find poor local optima in theory and practice.

We consider a different approach: we formulate system identification as an online learning problem, in which neither the data nor predictions are assumed to arise from an LDS. Furthermore, we slightly overparameterize the class of predictors, yielding an online convex program amenable to efficient regret minimization. This carefully chosen relaxation, which is our main theoretical contribution, expands the dimension of the hypothesis class by only a polylogarithmic factor. This construction relies upon recent work on the spectral theory of Hankel matrices.

The result is a simple and practical algorithm for time-series prediction, which deviates significantly from existing methods. We coin the term *wave-filtering* for our method, in reference to our relaxation's use of convolution by wave-shaped eigenvectors. We present experimental evidence on both toy data and a physical simulation, showing our method to be competitive in terms of predictive performance, more stable, and significantly faster than existing algorithms.

## 1.1 Our contributions

Consider a discrete-time linear dynamical system with inputs $\{x_t\}$, outputs $\{y_t\}$, and a latent state $\{h_t\}$, which can all be multi-dimensional. With noise vectors $\{\eta_t\}, \{\xi_t\}$, the system's time evolution is governed by the following equations:

$$h_{t+1} = Ah_t + Bx_t + \eta_t$$
$$y_t = Ch_t + Dx_t + \xi_t.$$

If the dynamics $A, B, C, D$ are known, then the Kalman filter [Kal60] is known to estimate the hidden state optimally under Gaussian noise, thereby producing optimal predictions of the system's response to any given input. However, this is rarely the case – indeed, real-world systems are seldom purely linear, and rarely are their evolution matrices known.

We henceforth give a provable, efficient algorithm for the prediction of sequences arising from an unknown dynamical system as above, in which the matrix $A$ is symmetric. Our main theoretical contribution is a regret bound for this algorithm, giving nearly-optimal convergence to the lowest mean squared prediction error (MSE) realizable by a symmetric LDS model:

**Theorem 1** (Main regret bound; informal)**.** *On an arbitrary sequence $\{(x_t, y_t)\}_{t=1}^T$, Algorithm 1 makes predictions $\{\hat{y}_t\}_{t=1}^T$ which satisfy*

$$\mathrm{MSE}(\hat{y}_1, \ldots, \hat{y}_T) - \mathrm{MSE}(\hat{y}_1^*, \ldots, \hat{y}_T^*) \; \leq \; \tilde{O}\left(\frac{\mathrm{poly}(n, m, d, \log T)}{\sqrt{T}}\right),$$

*compared to the best predictions $\{y_t^*\}_{t=1}^T$ by a symmetric LDS, while running in polynomial time.*

Note that the signal need not be generated by an LDS, and can even be *adversarially* chosen. In the less general batch (statistical) setting, we use the same techniques to obtain an analogous sample complexity bound for agnostic learning:

**Theorem 2** (Batch version; informal)**.** *For any choice of $\varepsilon > 0$, given access to an arbitrary distribution $\mathcal{D}$ over training sequences $\{(x_t, y_t)\}_{t=1}^T$, Algorithm 2, run on $N$ i.i.d. sample trajectories from $\mathcal{D}$, outputs a predictor $\hat{\Theta}$ such that*

$$\mathop{\mathbb{E}}_{\mathcal{D}}\left[\mathrm{MSE}(\hat{\Theta}) - \mathrm{MSE}(\Theta^*)\right] \; \leq \; \varepsilon + \frac{\tilde{O}\left(\mathrm{poly}(n, m, d, \log T, \log 1/\varepsilon)\right)}{\sqrt{N}},$$

*compared to the best symmetric LDS predictor $\Theta^*$, while running in polynomial time.*

Typical regression-based methods require the LDS to be *strictly* stable, and degrade on ill-conditioned systems; they depend on a spectral radius parameter $\frac{1}{1-\|A\|}$. Our proposed method of *wave-filtering* provably and empirically works even for the hardest case of $\|A\| = 1$. Our algorithm attains the first condition number-independent polynomial guarantees in terms of regret (equivalently, sample complexity) and running time for the MIMO setting. Interestingly, our algorithms never need to learn the hidden state, and our guarantees can be sharpened to handle the case when the dimensionality of $h_t$ is infinite.

## 1.2 Related work

The modern setting for LDS arose in the seminal work of Kalman [Kal60], who introduced the Kalman filter as a recursive least-squares solution for maximum likelihood estimation (MLE) of Gaussian perturbations to the system. The framework and filtering algorithm have proven to be a mainstay in control theory and time-series analysis; indeed, the term *Kalman filter model* is often used interchangeably with LDS. We refer the reader to the classic survey [Lju98], and the extensive overview of recent literature in [HMR16].

Ghahramani and Roweis [RG99] suggest using the EM algorithm to learn the parameters of an LDS. This approach, which directly tackles the non-convex problem, is widely used in practice [Mar10a]. However, it remains a long-standing challenge to characterize the theoretical guarantees afforded by EM. We find that it is easy to produce cases where EM fails to identify the correct system.

In a recent result of [HMR16], it is shown for the first time that for a restricted class of systems, gradient descent (also widely used in practice, perhaps better known in this setting as backpropagation)

guarantees polynomial convergence rates and sample complexity in the batch setting. Their result applies essentially only to the SISO case (vs. multi-dimensional for us), depends polynomially on the spectral gap (as opposed to no dependence for us), and requires the signal to be created by an LDS (vs. arbitrary for us).

## 2 Preliminaries

### 2.1 Linear dynamical systems

Many different settings have been considered, in which the definition of an LDS takes on many variants. We are interested in discrete time-invariant MIMO (multiple input, multiple output) systems with a finite-dimensional hidden state.[1] Formally, our model is given as follows:

**Definition 2.1.** *A linear dynamical system (LDS) is a map from a sequence of input vectors* $x_1, \ldots, x_T \in \mathbb{R}^n$ *to output (response) vectors* $y_1, \ldots, y_T \in \mathbb{R}^m$ *of the form*

$$h_{t+1} = Ah_t + Bx_t + \eta_t \tag{1}$$
$$y_t = Ch_t + Dx_t + \xi_t, \tag{2}$$

*where* $h_0, \ldots, h_T \in \mathbb{R}^d$ *is a sequence of hidden states,* $A, B, C, D$ *are matrices of appropriate dimension, and* $\eta_t \in \mathbb{R}^d, \xi_t \in \mathbb{R}^m$ *are (possibly stochastic) noise vectors.*

Unrolling this recursive definition gives the *impulse response function*, which uniquely determines the LDS. For notational convenience, for invalid indices $t \leq 0$, we define $x_t$, $\eta_t$, and $\xi_t$ to be the zero vector of appropriate dimension. Then, we have:

$$y_t = \sum_{i=1}^{T-1} CA^i \left(Bx_{t-i} + \eta_{t-i}\right) + CA^t h_0 + Dx_t + \xi_t. \tag{3}$$

We will consider the (discrete) time derivative of the impulse response function, given by expanding $y_{t-1} - y_t$ by Equation (3). For the rest of this paper, we focus our attention on systems subject to the following restrictions:

(i) The LDS is *Lyapunov stable*: $\|A\|_2 \leq 1$, where $\|\cdot\|_2$ denotes the operator (a.k.a. spectral) norm.

(ii) The transition matrix $A$ is symmetric and positive semidefinite.[2]

The first assumption is standard: when the hidden state is allowed to blow up exponentially, fine-grained prediction is futile. In fact, many algorithms only work when $\|A\|$ is *bounded away* from 1, so that the effect of any particular $x_t$ on the hidden state (and thus the output) dissipates exponentially. We do not require this stronger assumption.

We take a moment to justify assumption (ii), and why this class of systems is still expressive and useful. First, symmetric LDSs constitute a natural class of linearly-observable, linearly-controllable systems with dissipating hidden states (for example, physical systems with friction or heat diffusion). Second, this constraint has been used successfully for video classification and tactile recognition tasks [HSC$^+$16]. Interestingly, though our theorems require symmetric $A$, our algorithms appear to tolerate some non-symmetric (and even nonlinear) transitions in practice.

### 2.2 Sequence prediction as online regret minimization

A natural formulation of system identification is that of *online sequence prediction*. At each time step $t$, an online learner is given an input $x_t$, and must return a predicted output $\hat{y}_t$. Then, the true response $y_t$ is observed, and the predictor suffers a squared-norm loss of $\|y_t - \hat{y}_t\|^2$. Over $T$ rounds, the goal is to predict as accurately as the best LDS in hindsight.

Note that the learner is permitted to access the history of observed responses $\{y_1, \ldots, y_{t-1}\}$. Even in the presence of statistical (non-adversarial) noise, the fixed maximum-likelihood sequence produced by $\Theta = (A, B, C, D, h_0)$ will accumulate error linearly as $T$. Thus, we measure performance against a more powerful comparator, which fixes LDS parameters $\Theta$, and predicts $y_t$ by the previous response $y_{t-1}$ plus the derivative of the impulse response function of $\Theta$ at time $t$.

We will exhibit an online algorithm that can compete against the best $\Theta$ in this setting. Let $\hat{y}_1, \ldots, \hat{y}_T$ be the predictions made by an online learner, and let $y_1^*, \ldots, y_T^*$ be the sequence of predictions, realized by a chosen setting of LDS parameters $\Theta$, which minimize total squared error. Then, we define regret by the difference of total squared-error losses:

$$\text{Regret}(T) \stackrel{\text{def}}{=} \sum_{t=1}^{T} \|y_t - \hat{y}_t\|^2 - \sum_{t=1}^{T} \|y_t - y_t^*\|^2.$$

This setup fits into the standard setting of online convex optimization (in which a sublinear regret bound implies convergence towards optimal predictions), save for the fact that the loss functions are non-convex in the system parameters. Also, note that a randomized construction (set all $x_t = 0$, and let $y_t$ be i.i.d. Bernoulli random variables) yields a lower bound[3] for any online algorithm: $\mathbb{E}\left[\text{Regret}(T)\right] \geq \Omega(\sqrt{T})$.

To quantify regret bounds, we must state our scaling assumptions on the (otherwise adversarial) input and output sequences. We assume that the inputs are bounded: $\|x_t\|_2 \leq R_x$. Also, we assume that the output signal is Lipschitz in time: $\|y_t - y_{t-1}\|_2 \leq L_y$. The latter assumption exists to preclude pathological inputs where an online learner is forced to incur arbitrarily large regret. For a true noiseless LDS, $L_y$ is not too large; see Lemma F.5 in the appendix.

We note that an optimal $\tilde{O}(\sqrt{T})$ regret bound can be trivially achieved in this setting by algorithms such as Hedge [LW94], using an exponential-sized discretization of all possible LDS parameters; this is the online equivalent of brute-force grid search. Strikingly, our algorithms achieve essentially the same regret bound, but run in polynomial time.

## 2.3 The power of convex relaxations

Much work in system identification, including the EM method, is concerned with explicitly finding the LDS parameters $\Theta = (A, B, C, D, h_0)$ which best explain the data. However, it is evident from Equation 3 that the $CA^iB$ terms cause the least-squares (or any other) loss to be non-convex in $\Theta$. Many methods used in practice, including EM and subspace identification, heuristically estimate each hidden state $h_t$, after which estimating the parameters becomes a convex linear regression problem. However, this first step is far from guaranteed to work in theory or practice.

Instead, we follow the paradigm of improper learning: in order to predict sequences as accurately as the best possible LDS $\Theta^* \in \mathcal{H}$, one need not predict strictly from an LDS. The central driver of our algorithms is the construction of a slightly larger hypothesis class $\hat{\mathcal{H}}$, for which the best predictor $\hat{\Theta}^*$ is nearly as good as $\Theta^*$. Furthermore, we construct $\hat{\mathcal{H}}$ so that the loss functions *are* convex under this new parameterization. From this will follow our efficient online algorithm.

As a warmup example, consider the following overparameterization: pick some time window $\tau \ll T$, and let the predictions $\hat{y}_t$ be linear in the concatenation $[x_t, \ldots, x_{t-\tau}] \in \mathbb{R}^{\tau d}$. When $\|A\|$ is bounded away from 1, this is a sound assumption.[4] However, in general, this approximation is doomed to either truncate longer-term input-output dependences (short $\tau$), or suffer from overfitting (long $\tau$). Our main theorem uses an overparameterization whose approximation factor $\varepsilon$ is independent of $\|A\|$, and whose sample complexity scales only as $\tilde{O}(\text{polylog}(T, 1/\varepsilon))$.

## 2.4 Low approximate rank of Hankel matrices

Our analysis relies crucially on the spectrum of a certain *Hankel matrix*, a square matrix whose anti-diagonal stripes have equal entries (i.e. $H_{ij}$ is a function of $i + j$). An important example is the

Hilbert matrix $H_{n,\theta}$, the $n$-by-$n$ matrix whose $(i,j)$-th entry is $\frac{1}{i+j+\theta}$. For example,

$$H_{3,-1} = \begin{bmatrix} 1 & 1/2 & 1/3 \\ 1/2 & 1/3 & 1/4 \\ 1/3 & 1/4 & 1/5 \end{bmatrix}.$$

This and related matrices have been studied under various lenses for more than a century: see, e.g., [Hil94, Cho83]. A basic fact is that $H_{n,\theta}$ is a positive definite matrix for every $n \geq 1, \theta > -2$. The property we are most interested in is that the spectrum of a positive semidefinite Hankel matrix decays exponentially, a difficult result derived in [BT16] via Zolotarev rational approximations. We state these technical bounds in Appendix E.

## 3 The wave-filtering algorithm

Our online algorithm (Algorithm 1) runs online projected gradient descent [Zin03] on the squared loss $f_t(M_t) \overset{\text{def}}{=} \|y_t - \hat{y}_t(M_t)\|^2$. Here, each $M_t$ is a matrix specifying a linear map from featurized inputs $\tilde{X}_t$ to predictions $\hat{y}_t$. Specifically, after choosing a certain bank of $k$ *filters* $\{\phi_j\}$, $\tilde{X}_t \in \mathbb{R}^{nk+2n+m}$ consists of convolutions of the input time series with each $\phi_j$ (scaled by certain constants), along with $x_{t-1}, x_t$, and $y_{t-1}$. The number of filters $k$ will turn out to be polylogarithmic in $T$.

The filters $\{\phi_j\}$ and scaling factors $\{\sigma_j^{1/4}\}$ are given by the top eigenvectors and eigenvalues of the Hankel matrix $Z_T \in \mathbb{R}^{T \times T}$, whose entries are given by

$$Z_{ij} := \frac{2}{(i+j)^3 - (i+j)}.$$

In the language of Section 2.3, one should think of each $M_t$ as arising from an $\tilde{O}(\text{poly}(m,n,d,\log T))$-dimensional hypothesis class $\hat{\mathcal{H}}$, which replaces the original $O((m+n+d)^2)$-dimensional class $\mathcal{H}$ of LDS parameters $(A, B, C, D, h_0)$. Theorem 3 gives the key fact that $\hat{\mathcal{H}}$ approximately contains $\mathcal{H}$.

---

**Algorithm 1** Online wave-filtering algorithm for LDS sequence prediction

---

1: Input: time horizon $T$, filter parameter $k$, learning rate $\eta$, radius parameter $R_M$.
2: Compute $\{(\sigma_j, \phi_j)\}_{j=1}^k$, the top $k$ eigenpairs of $Z_T$.
3: Initialize $M_1 \in \mathbb{R}^{m \times k'}$, where $k' \overset{\text{def}}{=} nk + 2n + m$.
4: **for** $t = 1, \ldots, T$ **do**
5:     Compute $\tilde{X} \in \mathbb{R}^{k'}$, with first $nk$ entries $\tilde{X}_{(i,j)} := \sigma_j^{1/4} \sum_{u=1}^{T-1} \phi_j(u) x_{t-u}(i)$, followed by the $2n + m$ entries of $x_{t-1}, x_t$, and $y_{t-1}$.
6:     Predict $\hat{y}_t := M_t \tilde{X}$.
7:     Observe $y_t$. Suffer loss $\|y_t - \hat{y}_t\|^2$.
8:     Gradient update: $M_{t+1} \leftarrow M_t - 2\eta(y_t - \hat{y}_t) \otimes \tilde{X}$.
9:     **if** $\|M_{t+1}\|_F \geq R_M$ **then**
10:         Perform Frobenius norm projection: $M_{t+1} \leftarrow \frac{R_M}{\|M_{t+1}\|_F} M_{t+1}$.
11:     **end if**
12: **end for**

---

In Section 4, we provide the precise statement and proof of Theorem 1, the main regret bound for Algorithm 1, with some technical details deferred to the appendix. We also obtain analogous sample complexity results for batch learning; however, on account of some definitional subtleties, we defer all discussion of the offline case, including the statement and proof of Theorem 2, to Appendix A.

We make one final interesting note here, from which the name *wave-filtering* arises: when plotted coordinate-wise, our filters $\{\phi_j\}$ look like the vibrational modes of an inhomogeneous spring (see Figure 1). We provide some insight on this phenomenon (along with some other implementation concerns) in Appendix B. Succinctly: in the scaling limit, $(Z_T/\|Z_T\|_2)_{T \to \infty}$ commutes with a certain second-order Sturm-Liouville differential operator $\mathcal{D}$. This allows us to approximate filters with eigenfunctions of $\mathcal{D}$, using efficient numerical ODE solvers.

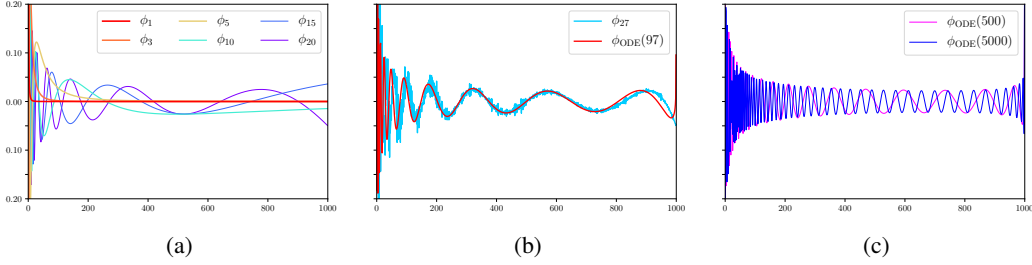

Figure 1: (a) The entries of some typical eigenvectors of $Z_{1000}$, plotted coordinate-wise. (b) $\phi_{27}$ of $Z_{1000}$ ($\sigma_{27} \approx 10^{-16}$) computed with finite-precision arithmetic, along with a numerical solution to the ODE in Appendix B.1 with $\lambda = 97$. (c) Some very high-order filters, computed using the ODE, would be difficult to obtain by eigenvector computations.

## 4 Analysis

We first state the full form of the regret bound achieved by Algorithm 1:[5]

**Theorem 1** (Main). *On any sequence $\{(x_t, y_t)\}_{t=1}^T$, Algorithm 1, with a choice of $k = \Theta\left(\log^2 T \log(R_\Theta R_x L_y n)\right)$, $R_M = \Theta(R_\Theta^2 \sqrt{k})$, and $\eta = \Theta((R_x^2 L_y \log(R_\Theta R_x L_y n) \, n\sqrt{T} \log^4 T)^{-1})$, achieves regret*

$$\text{Regret}(T) \leq O\left(R_\Theta^4 \, R_x^2 \, L_y \, \log^2(R_\Theta R_x L_y n) \cdot n\sqrt{T} \log^6 T\right),$$

*competing with LDS predictors $(A, B, C, D, h_0)$ with $0 \preccurlyeq A \preccurlyeq I$ and $\|B\|_F, \|C\|_F, \|D\|_F, \|h_0\| \leq R_\Theta$.*

Note that the dimensions $m, d$ do not appear explicitly in this bound, though they typically factor into $R_\Theta$. In Section 4.1, we state and prove Theorem 3, the convex relaxation guarantee for the filters, which may be of independent interest. This allows us to approximate the optimal LDS in hindsight (the regret comparator) by the loss-minimizing matrix $M_t : \tilde{X} \mapsto \hat{y}_t$. In Section 4.2, we complete the regret analysis using Theorem 3, along with bounds on the diameter and gradient, to conclude Theorem 1.

Since the batch analogue is less general (and uses the same ideas), we defer discussion of Algorithm 2 and Theorem 2 to Appendix A.

### 4.1 Approximate convex relaxation via wave filters

Assume for now that $h_0 = 0$; we will remove this at the end, and see that the regret bound is asymptotically the same. Recall (from Section 2.2) that we measure regret compared to predictions obtained by adding the derivative of the impulse response function of an LDS $\Theta$ to $y_{t-1}$. Our approximation theorem states that for any $\Theta$, there is some $M_\Theta \in \hat{\mathcal{H}}$ which produces approximately the same predictions. Formally:

**Theorem 3** (Spectral convex relaxation for symmetric LDSs). *Let $\{\hat{y}_t\}_{t=1}^T$ be the online predictions made by an LDS $\Theta = (A, B, C, D, h_0 = 0)$. Let $R_\Theta = \max\{\|B\|_F, \|C\|_F, \|D\|_F\}$. Then, for any $\varepsilon > 0$, with a choice of $k = \Omega\left(\log T \log(R_\Theta R_x L_y nT/\varepsilon)\right)$, there exists an $M_\Theta \in \mathbb{R}^{m \times k'}$ such that*

$$\sum_{t=1}^T \|M_\Theta \tilde{X}_t - y_t\|^2 \leq \sum_{t=1}^T \|\hat{y}_t - y_t\|^2 + \varepsilon.$$

*Here, $k'$ and $\tilde{X}_t$ are defined as in Algorithm 1 (noting that $\tilde{X}_t$ includes the previous ground truth $y_{t-1}$).*

*Proof.* We construct this mapping $\Theta \mapsto M_\Theta$ explicitly. Write $M_\Theta$ as the block matrix

$$\begin{bmatrix} M^{(1)} & M^{(2)} & \cdots & M^{(k)} & M^{(x')} & M^{(x)} & M^{(y)} \end{bmatrix},$$

where the blocks' dimensions are chosen to align with $\tilde{X}_t$, the concatenated vector

$$\begin{bmatrix} \sigma_1^{1/4}(X * \phi_1)_t & \sigma_2^{1/4}(X * \phi_2)_t & \cdots & \sigma_k^{1/4}(X * \phi_k)_t & x_{t-1} & x_t & y_{t-1} \end{bmatrix},$$

so that the prediction is the block matrix-vector product

$$M_\Theta \tilde{X}_t = \sum_{j=1}^k \sigma_j^{1/4} M^{(j)} (X * \phi_j)_t + M^{(x')} x_{t-1} + M^{(x)} x_t + M^{(y)} y_{t-1}.$$

Without loss of generality, assume that $A$ is diagonal, with entries $\{\alpha_l\}_{l=1}^d$.[6] Let $b_l$ be the $l$-th row of $B$, and $c_l$ the $l$-th column of $C$. Also, we define a continuous family of vectors $\mu : [0,1] \to \mathbb{R}^T$, with entries $\mu(\alpha)(i) = (\alpha_l - 1)\alpha_l^{i-1}$. Then, our construction is as follows:

- $M^{(j)} = \sum_{l=1}^d \sigma_j^{-1/4} \langle \phi_j, \mu(\alpha_l) \rangle (c_l \otimes b_l)$, for each $1 \le j \le k$.

- $M^{(x')} = -D, \quad M^{(x)} = CB + D, \quad M^{(y)} = I_{m \times m}.$

Below, we give the main ideas for why this $M_\Theta$ works, leaving the full proof to Appendix C.

Since $M^{(y)}$ is the identity, the online learner's task is to predict the differences $y_t - y_{t-1}$ as well as the derivative $\Theta$, which we write here:

$$\hat{y}_t - y_{t-1} = (CB + D)x_t - Dx_{t-1} + \sum_{i=1}^{T-1} C(A^i - A^{i-1})Bx_{t-i}$$

$$= (CB + D)x_t - Dx_{t-1} + \sum_{i=1}^{T-1} C \left( \sum_{l=1}^d \left(\alpha_l^i - \alpha_l^{i-1}\right) e_l \otimes e_l \right) Bx_{t-i}$$

$$= (CB + D)x_t - Dx_{t-1} + \sum_{l=1}^d (c_l \otimes b_l) \sum_{i=1}^{T-1} \mu(\alpha_l)(i) \, x_{t-i}. \tag{4}$$

Notice that the inner sum is an inner product between each coordinate of the past inputs $(x_t, x_{t-1}, \ldots, x_{t-T})$ with $\mu(\alpha_l)$ (or a convolution, viewed across the entire time horizon). The crux of our proof is that one can approximate $\mu(\alpha)$ using a linear combination of the filters $\{\phi_j\}_{j=1}^k$. Writing $Z := Z_T$ for short, notice that

$$Z = \int_0^1 \mu(\alpha) \otimes \mu(\alpha) \, d\alpha,$$

since the $(i,j)$ entry of the RHS is

$$\int_0^1 (\alpha - 1)^2 \alpha^{i+j-2} \, d\alpha = \frac{1}{i+j-1} - \frac{2}{i+j} + \frac{1}{i+j+1} = Z_{ij}.$$

What follows is a spectral bound for reconstruction error, relying on the low approximate rank of $Z$:

**Lemma 4.1.** *Choose any $\alpha \in [0,1]$. Let $\tilde{\mu}(\alpha)$ be the projection of $\mu(\alpha)$ onto the $k$-dimensional subspace of $\mathbb{R}^T$ spanned by $\{\phi_j\}_{j=1}^k$. Then,*

$$\|\mu(\alpha) - \tilde{\mu}(\alpha)\|^2 \le \sqrt{6 \sum_{j=k+1}^T \sigma_j} \le O\left( c_0^{-k/\log T} \sqrt{\log T} \right),$$

*for an absolute constant $c_0 > 3.4$.*

By construction of $M^{(j)}$, $M_\Theta \tilde{X}_t$ replaces each $\mu(\alpha_l)$ in Equation (4) with its approximation $\tilde{\mu}(\alpha_l)$. Hence we conclude that

$$M_\Theta \tilde{X}_t = y_{t-1} + (CB + D)x_t - Dx_{t-1} + \sum_{l=1}^{d}(c_l \otimes b_l) \sum_{i=1}^{T-1} \tilde{\mu}(\alpha_l)(i)\, x_{t-i}$$

$$= y_{t-1} + (\hat{y}_t - y_{t-1}) + \zeta_t \quad = \hat{y}_t + \zeta_t,$$

letting $\{\zeta_t\}$ denote some residual vectors arising from discarding the subspace of dimension $T - k$. Theorem 3 follows by showing that these residuals are small, using Lemma 4.1: it turns out that $\|\zeta_t\|$ is exponentially small in $k/\log T$, which implies the theorem. $\qquad\square$

## 4.2  From approximate relaxation to low regret

Let $\Theta^* \in \mathcal{H}$ denote the best LDS predictor, and let $M_{\Theta^*} \in \hat{\mathcal{H}}$ be its image under the map from Theorem 3, so that total squared error of predictions $M_{\Theta^*}\tilde{X}_t$ is within $\varepsilon$ from that of $\Theta^*$. Notice that the loss functions $f_t(M) \overset{\text{def}}{=} \|y_t - M\tilde{X}_t\|^2$ are quadratic in $M$, and thus convex. Algorithm 1 runs online gradient descent [Zin03] on these loss functions, with decision set $\mathcal{M} \overset{\text{def}}{=} \{M \in \mathbb{R}^{m \times k'} \mid \|M\|_F \leq R_M\}$. Let $D_{\max} := \sup_{M,M' \in \mathcal{M}}\|M - M'\|_F$ be the diameter of $\mathcal{M}$, and $G_{\max} := \sup_{M \in \mathcal{M}, \tilde{X}}\|\nabla f_t(M)\|_F$ be the largest norm of a gradient. We can invoke the classic regret bound:

**Lemma 4.2** (e.g. Thm. 3.1 in [Haz16]). *Online gradient descent, using learning rate $\frac{D_{\max}}{G_{\max}\sqrt{T}}$, has regret*

$$\text{Regret}_{\text{OGD}}(T) \overset{\text{def}}{=} \sum_{t=1}^{T} f_t(M_t) - \min_{M \in \mathcal{M}} \sum_{t=1}^{T} f_t(M) \leq 2G_{\max}D_{\max}\sqrt{T}.$$

To finish, it remains to show that $D_{\max}$ and $G_{\max}$ are small. In particular, since the gradients contain convolutions of the input by $\ell_2$ (not $\ell_1$) unit vectors, special care must be taken to ensure that these do not grow too quickly. These bounds are shown in Section D.2, giving the correct regret of Algorithm 1 in comparison with the comparator $M^* \in \hat{\mathcal{H}}$. By Theorem 3, $M^*$ competes arbitrarily closely with the best LDS in hindsight, concluding the theorem.

Finally, we discuss why it is possible to relax the earlier assumption $h_0 = 0$ on the initial hidden state. Intuitively, as more of the ground truth responses $\{y_t\}$ are revealed, the largest possible effect of the initial state decays. Concretely, in Section D.4, we prove that a comparator who chooses a nonzero $h_0$ can only increase the regret by an additive $\tilde{O}(\log^2 T)$ in the online setting.

# 5  Experiments

In this section, to highlight the appeal of our provable method, we exhibit two minimalistic cases where traditional methods for system identification fail, while ours successfully learns the system. Finally, we note empirically that our method seems not to degrade in practice on certain well-behaved nonlinear systems. In each case, we use $k = 25$ filters, and a regularized follow-the-leader variant of Algorithm 1 (see Appendix B.2).

## 5.1  Synthetic systems: two hard cases for EM and SSID

We construct two difficult systems, on which we run either EM or subspace identification[7] (SSID), followed by Kalman filtering to obtain predictions. Note that our method runs significantly (>1000 times) faster than this traditional pipeline.

In the first example (Figure 2(a), left), we have a SISO system ($n = m = 1$) and $d = 2$; all $x_t$, $\xi_t$, and $\eta_t$ are i.i.d. Gaussians, and $B^\top = C = [1\ \ 1]$, $D = 0$. Most importantly, $A = \text{diag}\,([0.999, 0.5])$ is ill-conditioned, so that there are long-term dependences between input and output. Observe that although EM and SSID both find reasonable guesses for the system's dynamics, they turns out to be local optima. Our method learns to predict as well as the *best possible* LDS.

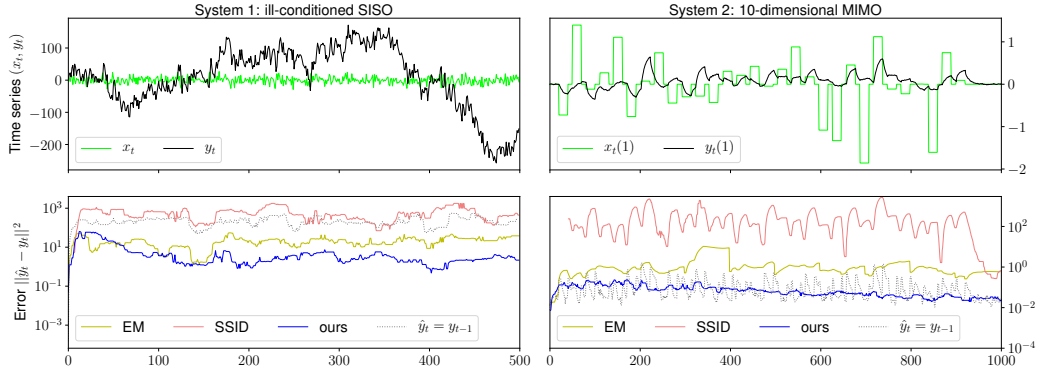

(a) Two synthetic systems. For clarity, error plots are smoothed by a median filter. *Left:* Noisy SISO system with a high condition number; EM and SSID finds a bad local optimum. *Right:* High-dimensional MIMO system; other methods fail to learn any reasonable model of the dynamics.

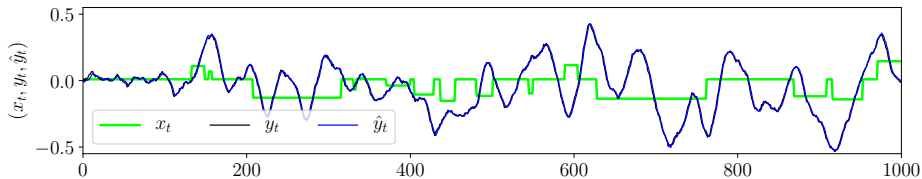

(b) Forced pendulum, a physical simulation our method learns in practice, despite a lack of theory.

Figure 2: Visualizations of Algorithm 1. All plots: blue = ours, yellow = EM, red = SSID, **black** = true responses, green = inputs, dotted lines = "guess the previous output" baseline. Horizontal axis is time.

The second example (Figure 2(a), right) is a MIMO system (with $n = m = d = 10$), also with Gaussian noise. The transition matrix $A = \mathrm{diag}\,([0, 0.1, 0.2, \dots, 0.9])$ has a diverse spectrum, the observation matrix $C$ has i.i.d. Gaussian entries, and $B = I_n, D = 0$. The inputs $x_t$ are random block impulses. This system identification problem is high-dimensional and non-convex; it is thus no surprise that EM and SSID consistently fail to converge.

## 5.2   The forced pendulum: a nonlinear, non-symmetric system

We remark that although our algorithm has provable regret guarantees only for LDSs with symmetric transition matrices, it appears in experiments to succeed in learning some non-symmetric (even nonlinear) systems in practice, much like the unscented Kalman filter [WVDM00]. In Figure 2(b), we provide a typical learning trajectory for a forced pendulum, under Gaussian noise and random block impulses. Physical systems like this are widely considered in control and robotics, suggesting possible real-world applicability for our method.

## 6   Conclusion

We have proposed a novel approach for provably and efficiently learning linear dynamical systems. Our online *wave-filtering* algorithm attains near-optimal regret in theory; and experimentally out-performs traditional system identification in both prediction quality and running time. Furthermore, we have introduced a "spectral filtering" technique for convex relaxation, which uses convolutions by eigenvectors of a Hankel matrix. We hope that this theoretical tool will be useful in tackling more general cases, as well as other non-convex learning problems.

## Acknowledgments

We thank Holden Lee and Yi Zhang for helpful discussions. We especially grateful to Holden for a thorough reading of our manuscript, and for pointing out a way to tighten the result in Lemma C.1.

## Footnotes

[1]We assume finite dimension for simplicity of presentation. However, it will be evident that hidden-state dimension has no role in our algorithm, and shows up as $\|B\|_F$ and $\|C\|_F$ in the regret bound.

[2]The psd constraint on $A$ can be removed by augmenting the inputs $x_t$ with extra coordinates $(-1)^t(x_t)$. We omit this for simplicity of presentation.

[3]This is a standard construction; see, e.g. Theorem 3.2 in [Haz16].

[4]This assumption is used in *autoregressive models*; see Section 6 of [HMR16] for a theoretical treatment.

[5]Actually, for a slightly tighter proof, we analyze a restriction of the algorithm which does not learn the portion $M^{(y)}$, instead always choosing the identity matrix for that block.

[6]Write the eigendecomposition $A = U\Lambda U^T$. Then, the LDS with parameters $(\hat{A}, \hat{B}, \hat{C}, D, h_0) := (\Lambda, BU, U^T C, D, h_0)$ makes the same predictions as the original, with $\hat{A}$ diagonal.

[7]Specifically, we use "Deterministic Algorithm 1" from page 52 of [VODM12].

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
