[Supplementary Material]

# Supplementary Material for "Learning Linear Dynamical Systems via Spectral Filtering"

**Elad Hazan, Karan Singh, Cyril Zhang**
Department of Computer Science
Princeton University
Princeton, NJ 08544
{ehazan,karans,cyril.zhang}@cs.princeton.edu

## Guide to the Appendix

- In Appendix A, we present two formulations of the batch learning equivalent of the online algorithm, and derive Theorem 2, a companion sample complexity bound.

- In Appendix B, we discuss some variants of our online algorithm, and offer some tips for implementation. We also provide discussion on the connection of our filters to eigenfunctions of a certain differential operator.

- In Appendix C, we prove the key approximate convex relaxation result (Theorem 3).

- In Appendix D, we complete the details for the proof sketch provided in Section 4.2, concluding the main theorem, the regret bound for the online algorithm. Importantly, we address the subtle issue of deriving upper bounds for the gradient and diameter of the decision set.

- In Appendix E, we derive explicit non-asymptotic bounds for quantities of interest pertaining to the Hankel matrix $Z$, notably spectral decay. Key results are adapted from [BT16].

- In Appendix F, we verify some easy-to-prove properties of the important vector $\mu(\alpha)$, for sake of completeness.

## A    Batch variants of the algorithm

The online prediction setting is sensitive to permutation of the time series: that is, the same LDS does not in general map $\{x_{\sigma(1)}, \ldots, x_{\sigma(T)}\}$ to $\{y_{\sigma(1)}, \ldots, y_{\sigma(T)}\}$. As such, one must take care when defining the batch case: the output time series (and thus, loss functions) are correlated, so it is not meaningful to assume that they are i.i.d. samples from a distribution. Thus, our online regret bound, which concerns a single episode, does not translate directly. However, our convex relaxation technique still allows us to do efficient improper learning with least-squares regression, giving interesting and novel statistical guarantees. In this section, we provide two possible formulations of the batch setting, along with accompanying theorems.

In both cases, it is most natural to fix an *episode length* $T$, and consider a rollout of the system $\{(x_t, y_t)\}_{t=1}^{T}$ to be a single example. For short, let $X_i \in \mathbb{R}^{Tn}$ denote the concatenated vector of inputs for a single example, and $Y_i \in \mathbb{R}^{Tm}$ the concatenated responses. The batch formulation is to learn the dynamics of the system using $N$ samples $\{(X_i, Y_i)\}$. Recall that the samples satisfy $\|x_t\|_2 \leq R_x$ and $\|y_t - y_{t-1}\|_2 \leq L_y$.

Unlike in the online setting, it will be less confusing in the batch setting to measure the *mean* squared error of predictions, rather than the total squared error. Thus, in this section, $\ell_{X,Y}(h)$ will always refer to mean squared error. As well, to follow statistical learning conventions (for ease of reading),

we use $h$ to denote a hypothesis (an LDS) instead of $\Theta$; this is distinguished from the hidden state $h_t$.

## A.1 Learning the derivative: the direct analogue

Throughout this subsection, assume that $h_0 = 0$.

As noted, the sequential prediction algorithm can be restricted so as to never make updates to the submatrix $M^{(y)}$, keeping it to be the identity matrix. Notice that all other features in $\tilde{X}$ consist of inputs $x_t$ and their convolutions. In other words, we can take the view that the matrix $M_t$ can be used to predict the *differences* $y_t - y_{t-1}$ between successive responses, as a function of the entire (aligned) input time series $(x_t, x_{t-1}, \ldots, x_{t_T})$.

Thus, we can formulate a direct analogue for the online algorithm: learn the mapping from an input time series $X_i \in \mathbb{R}^{Tn}$ to the *differences* $Y_i' \in \mathbb{R}^{Tm}$, the concatenation of all $y_t - y_{t-1}$. For this, we can use Theorem 3 (the approximation result) directly, and obtain an improper agnostic learning guarantee.

Specifically, let $\mathcal{H}$ be a subset of the hypothesis class of LDS parameters $\Theta = (A, B, C, D, h_0 = 0)$, subject to $\|B\|_F, \|C\|_F, \|D\|_F \leq R_\Theta$, and choose any approximation tolerance $\varepsilon > 0$.[1] Then, Theorem 3 states that choosing $\hat{\mathcal{H}}$ with $k = \Omega\left(\log T \log(R_\Theta R_x L_y nT/\varepsilon)\right)$ ensures the $\varepsilon$-approximate relaxation property. In the language of the batch setting: for each $h \in \mathcal{H}$ which predicts on the sample $(X, Y')$ with a mean squared error $\ell_X(h)$, there is some $\hat{h} \in \hat{\mathcal{H}}$ so that

$$\ell_{X,Y}(h) \leq \ell_{X,Y}(\hat{h}) + \varepsilon.$$

The choice of batch algorithm is clear, in order to mimic Algorithm 1: run least-squares regression on $\tilde{X}$ and $Y$, where $\tilde{X}$ is the same featurization of the inputs as used in the online algorithm. We describe this procedure fully in Algorithm 2.

---

**Algorithm 2** Offline wave-filtering algorithm for learning the derivative of an LDS

---

1: **Input:** $S = \{(X_i, Y_i')\}$, a set of $N$ training samples, each of length $T$; filter parameter $k$.
2: Compute $\{(\sigma_j, \phi_j)\}_{j=1}^k$, the top $k$ eigenpairs of $Z_T$.
3: Initialize matrices $\mathbf{X} \in \mathbb{R}^{(nk+2n) \times NT}$, $\mathbf{Y}' \in \mathbb{R}^{m \times NT}$.
4: **for** each sample $(X, Y')$ **do**
5:     **for** $t = 1, \ldots, T$ **do**
6:         Compute $\tilde{X}_t \in \mathbb{R}^{nk+2n}$, with first $nk$ entries $\tilde{X}_{(i,j)} := \sigma_j^{1/4} \sum_{u=1}^{T-1} \phi_j(u) x_{t-u}(i)$, followed by the $2n$ entries of $x_{t-1}, x_t$.
7:         Append $(\tilde{X}_t, Y_t')$ as new columns to the matrices $\mathbf{X}, \mathbf{Y}'$.
8:     **end for**
9: **end for**
10: **return** least-squares solution $(\mathbf{X}\mathbf{X}^\top)^\dagger \mathbf{X}^\top \mathbf{Y}'$.

---

### A.1.1 Generalization bound

By definition, Algorithm 2 minimizes the empirical MSE loss on the samples; as such, we can derive a PAC-learning bound for regression. We begin with some definitions and assumptions, so that we can state the theorem.

As in the statement of the online algorithm, as a soft dimensionality restriction, we constrain the comparator class $\mathcal{H}$ to contain LDSs with parameters $\Theta = (A, B, C, D, h_0 = 0)$ such that $0 \preccurlyeq A \preccurlyeq I$ and $\|B\|_F, \|C\|_F, \|D\|_F, \|h_0\| \leq R_\Theta$. For an empirical sample set $S$, let $\ell_S(h) = \frac{1}{|S|} \sum_{(X,Y) \in S} \ell_{X,Y}(h)$. Similarly, for a distribution $\mathcal{D}$, let $\ell_\mathcal{D}(h) = \mathbb{E}_{(X,Y) \sim \mathcal{D}}[\ell_{X,Y}(h)]$.

Then, we are able to obtain a sample complexity bound:

**Theorem 2** (Generalization of the batch algorithm). *Choose any $\varepsilon > 0$. Let $S = \{(X_i, Y_i')\}_{i=1}^N$ be a set of i.i.d. training samples from a distribution $\mathcal{D}$. Let $\hat{h} \stackrel{def}{=} \operatorname{argmin}_{h \in \hat{\mathcal{H}}} \ell_S(h)$ be the output of Algorithm 2, with a choice of $k = \Theta(\log T \; \log(R_\Theta R_x L_y n T / \varepsilon))$. Let $h^* \stackrel{def}{=} \operatorname{argmin}_{h^* \in \mathcal{H}} \ell_\mathcal{D}(h)$ be the true loss minimizer. Then, with probability at least $1 - \delta$, it holds that*

$$\ell_\mathcal{D}(\hat{h}) - \min_{h \in \mathcal{H}} \ell_\mathcal{D}(h) \leq \varepsilon + \frac{O\left(R_\Theta^4 R_x^2 L_y \, \log^2(R_\Theta R_x L_y n / \varepsilon) \, n \log^6 T + \sqrt{\log 1/\delta}\right)}{\sqrt{N}}.$$

*Proof.* Lemma D.1 shows that we can restrict $\hat{\mathcal{H}}$ by a Frobenius norm bound:

$$\|M\|_F \leq O\left(R_\Theta^2 \sqrt{k}\right).$$

Thus, the empirical Rademacher complexity of $\hat{\mathcal{H}}$ on $N$ samples, with this restriction, thus satisfies

$$\mathcal{R}_N(\hat{\mathcal{H}}) \leq O\left(\frac{R_\Theta^2 R_x \sqrt{k}}{\sqrt{N}}\right).$$

Also, no single prediction error (and thus neither the empirical nor population loss) will exceed the upper bound

$$\ell_{\max} \stackrel{def}{=} \Theta(R_\Theta^4 R_x^2 L_y^2 k).$$

Finally, the loss is $G_{\max}$-Lipschitz in the matrix $h$, where $G_{\max}$ is the same upper bound for the gradient as mentioned in Section 4.2. Lemma D.5, states that this is bounded by $O\left(R_\Theta^2 R_x^2 L_y \cdot n k^{3/2} \log^2 T\right)$.

With all of these facts in hand, a standard Rademacher complexity-dependent generalization bound holds in the improper hypothesis class $\hat{\mathcal{H}}$ (see, e.g. [BM02]):

**Lemma A.1** (Generalization via Rademacher complexity). *With probability at least $1 - \delta$, it holds that*

$$\ell_\mathcal{D}(\hat{h}) - \ell_\mathcal{D}(\hat{h}^*) \leq G_{\max} \mathcal{R}_N(\hat{\mathcal{H}}) + \ell_{\max} \sqrt{\frac{8 \ln 2/\delta}{N}}$$

With the stated choice of $k$, an upper bound for the RHS of Lemma A.1 is

$$\frac{O\left(R_\Theta^4 R_x^2 L_y \, \log^2(R_\Theta R_x L_y n / \varepsilon) \, n \log^6 T + \sqrt{\log 1/\delta}\right)}{\sqrt{N}}.$$

Combining this with the approximation result (Theorem 3) yields the theorem. $\square$

### A.2 The pure batch setting

A natural question is whether there exists a batch learning algorithm that can use $X$ to predict $Y$ directly, as opposed to the differences $Y'$. This is possible in the regime of low noise: if one has predictions on $Y'$ that are correct up to MSE $\varepsilon$, an easy solution is to integrate and obtain predictions for $Y$; however, the errors will accumulate to $T\varepsilon$. The same agnostic learning guarantee costs a rather dramatic factor of $T^2$ in sample complexity.

In the regime of low noise, an analogue of our approximation theorem (Theorem 3) is powerful enough to guarantee low error. For convenience and concreteness, we record this here:

**Theorem 3b** (Pure-batch approximation). *Let $\Theta$ be an LDS specified by parameters $(A, B, C, D, h_0 = 0)$, with $0 \preccurlyeq A \preccurlyeq I$, and $\|B\|_F, \|C\|_F, \|D\|_F \leq R_\Theta$. Suppose $\Theta$ takes an input sequence $X = \{x_1, \ldots, x_T\}$, and produces output sequence $Y = \{y_1, \ldots, y_T\}$, assuming all noise vectors $\xi_t, \eta_t$ are 0. Then, for any $\varepsilon > 0$, with a choice of $k = \Omega\left(\log T \log(R_\Theta R_x L_y n T / \varepsilon)\right)$, there exists an $M_\Theta \in \mathbb{R}^{m \times (nk+2n)}$ such that*

$$\sum_{t=1}^T \left\| \left(\sum_{u=1}^t M_\Theta \tilde{X}_u\right) - y_t \right\|^2 \leq \sum_{t=1}^T \|\hat{y}_t - y_t\|^2 + \varepsilon,$$

*where $\tilde{X}_t$ is defined as in Algorithm 1, without the $y_{t-1}$ entries.*

This fact follows from Theorem 3, setting $\varepsilon/T$ as the desired precision; the cost of this additional precision is only a constant factor in $k$. Furthermore, this $M_\Theta$ is subject to the same Frobenius norm constraint $\|M_\Theta\|_F \leq O(R_\Theta^2 \sqrt{k})$ as in Lemma D.1.

### A.2.1 Filters from the Hilbert matrix

Alternatively, in the realizable case (when the samples from $\mathcal{D}$ are generated by an LDS, possibly with small noise), one can invoke a similar approximate relaxation theorem as Theorem 3. The filters become the eigenvectors of the Hilbert matrix $H_{T,-1}$, the matrix whose $(i,j)$-th entry is $1/(i+j-1)$. This matrix exhibits the same spectral decay as $Z_T$; see [BT16] for precise statements. the proof follows the sketch from Section 4.1, approximating the powers of $\alpha_\ell$ by a spectral truncation of a different curve $\mu'(\alpha)(i) = \alpha^{i-1}$, sometimes called the *moment curve* in $\mathbb{R}^T$. The Hilbert matrix arises from taking the second moment matrix of the uniform distribution on this curve.

However, we find that this approximation guarantee is insufficient to show the strong regret and agnostic learning bounds we exhibit for learning the derivative of the impulse response function. Nonetheless, we find that regression with these filters works well in practice, even interchangeably in the online algorithm; see Section B.1 for some intuition.

### A.3 Learning the initial hidden state via hints

In either of the above settings, it is not quite possible to apply the same argument as in the online setting for pretending that the initial hidden state is zero. When this assumption is removed, the quality of the convex relaxation degrades by an *additive* $\tilde{O}(\frac{\log^2 T}{T})$; see Section D.4. This does not matter much for the regret bound, because it is subsumed by the worst-case regret of online gradient descent.

However, in the batch setting, we take the view of fixed $T$ and increasing $N$, so the contribution of the initial state is no longer asymptotically negligible. In other words, this additive approximation error hinders us from driving $\varepsilon$ arbitrarily close to zero, no matter how many filters are selected. In settings where $T$ is large enough, one may find this acceptable.

We present an augmented learning problem in which we *can* predict as well as an LDS: the initial hidden state is provided in each sample, up to an arbitrary linear transformation. Thus, each sample takes the form $(X, Y, \tilde{h}_0)$, and it is guaranteed that $\tilde{h}_0 = Q h_0$ for each sample, for a fixed matrix $Q \in \mathbb{R}^{d' \times d}$. This $Q$ must be well-conditioned for the problem to remain well-posed: our knowledge of $h_0$ should be in the same dynamic range as the ground truth. Concretely, we should assume that $\sigma_{\max}(Q)/\sigma_{\min}(Q)$ is bounded.

The construction is as follows: append $d'$ "dummy" dimensions to the input, and add an impulse of $\tilde{h}_0$ in those dimensions at time 0. During the actual episode, these dummy inputs are always zero. Then, replacing $B$ with the augmented block matrix $\begin{bmatrix} B & Q^{-1} \end{bmatrix}$ recovers the behavior of the system. Thus, we can handle this formulation of hidden-state learning in the online or batch setting, incurring no additional asymptotic factors.

### A.3.1 Initializations with finite support

We highlight an important special case of the formulation discussed above, which is perhaps the motivating rationale for this altered problem.

Consider a batch system identification setting in which there are only *finitely many* initial states $h_0$ in the training and test data, and the experimenter can distinguish between these states. This can be interpreted a set of $n_{\text{hidden}}$ known initial "configurations" of the system. Then, it is sufficient to augment the data with a one-hot vector in $\mathbb{R}^{n_{\text{hidden}}}$, corresponding to the known initialization in each sample. An important case is when $n_{\text{hidden}} = 1$: when there is only *one* distinct initial configuration; this occurs frequently in control problems.

In summary, the stated augmentation takes the original LDS with dimensions $(n, m, d, T)$, and transforms it into one with dimensions $(n + n_{\text{hidden}}, m, d, T + 1)$. The matrix $Q^{-1}$, as defined above, is the $n_{\text{hidden}}$-by-$d$ matrix whose columns are the possible initial hidden states, which can be in arbitrary dimension. For convenience, we summarize this observation:

**Proposition A.2** (Learning an LDS with few, distinguishable hidden states). *When there are $d'$ known hidden states, with $d'\|h_0\| \leq R_\Theta$, Theorems 2, 3, and 3b apply to the modified LDS learning problem, with samples of the form $(\tilde{h}_0, X, Y)$. The dimension $n$ becomes $n + d'$.*

# B  Implementation and variants

We discuss the points mentioned in Section 3 at greater length. Unlike the rest of the appendix, this section contains no technical proofs, and is intended as a user-friendly guide for making the wave-filtering method usable in practice.

## B.1  Computing the filters via Sturm-Liouville ODEs

We begin by expanding upon the observation, noted in Section 3, that the eigenvectors resemble inhomogeneously-oscillating waves, providing some justification for the heuristic numerical computation of the top eigenvectors of $Z_T$.

Computing the filters directly from $Z_T$ is difficult. In fact, the Hilbert matrix (its close cousin) is notoriously super-exponentially ill-conditioned; it is probably best known for being a pathological benchmark for finite-precision numerical linear algebra algorithms. One could ignore efficiency issues, and view this as a data-independent preprocessing step: these filters are deterministic. However, this difficult numerical problem poses an issue for using our method in practice.

Fortunately, as briefly noted in Section 3, some recourse is available. In [Grü82], Grünbaum constructs a tridiagonal matrix $T_{n,\theta}$ which commutes with each Hilbert matrix $H_{n,\theta}$, as defined in Section 2.4. In the appropriate scaling limit as $T \to \infty$, this $T_{n,\theta}$ becomes a Sturm-Liouville differential operator $\mathcal{D}$ which does not depend on $\theta$, given by

$$\mathcal{D} = \frac{d}{dx}\left((1 - x^2)x^2 \frac{d}{dx}\right) - 2x^2.$$

Notice that $Z_T = H_{T,-1} - 2H_{T,0} + H_{T,1}$. This suggests that for large $T$, the entries of the $\phi_j$ are approximated by solutions to the second-order ODE

$$\mathcal{D}\phi = \lambda\phi. \tag{1}$$

It is difficult to quantify theoretical bounds for this rather convoluted sequence of approximations; however, we find that this observation greatly aids with constructing these filters in practice. In particular, the map between eigenvalues $\sigma_j$ of $Z$ and $\lambda_j$ of $\mathcal{D}$ corresponding to the same eigenvector/eigenfunction proves challenging to characterize for finite $T$. In practice, we find that our method's performance is sensitive to neither the precise eigenvalues nor the ODE boundary conditions.

In summary, aside from the name *wave-filtering*, this observation yields a numerically stable recipe for computing filters (without a theorem): for each of $k$ hand-selected eigenvalues $\lambda$, compute a filter $\phi_\lambda$ using an efficient numerical solver to Equation 1.

## B.2  Alternative low-regret algorithms

We use online gradient descent as our prototypical low-regret learning algorithm due to its simplicity and stability under worst-case noise. However, in practice, particularly when there are additional structural assumptions on the data, we can replace the update step with that of any low-regret algorithm. AdaGrad [DHS11] is a particularly appealing one, as it is likely to find learning rates which are better than those guaranteed theoretically.

Furthermore, if noise levels are relatively low, and it is known *a priori* that the data are generated from a true LDS, a better approach might be to use follow-the-leader [KV05] or any of its variants. This amounts to replacing the update step with

$$M_{t+1} := \min_M \sum_{t'=1}^{t} \|y_{t'} - \hat{y}_{t'}(M)\|^2,$$

a linear regression problem solvable via, e.g. conjugate gradient. For such iterative methods, we further note that it is possible to use the previous predictor $M_{t-1}$ as a warm start.

### B.3 Accelerating convolutions

In the batch setting (or in the online setting, when all the inputs $x_t$ are known in advance), it is easy to see that the convolution components of all feature vectors $\tilde{X}_t$ can be computed in a single pass, by pointwise multiplication in the Fourier domain. Using the fast Fourier transform, one can implement all convolutions in time $O(nkT \log T)$, nearly linear in the size of the input. This mitigates what would otherwise be a quadratic dependence on $T$. Many software libraries provide an FFT-based implementation of convolution.

## C  Proof of the relaxation theorem

In this section, we follow the proof structure given in Section 4.1, and conclude Theorem 3.

Before proceeding, we note here that the algorithm could have used filters of length $T-1$ instead of $T$, obtained from the eigenvectors of $Z_{T-1}$. However, since carrying this $-1$ through the statements and analysis degrades clarity significantly, we use a slightly suboptimal matrix throughout this exposition.

### C.1  Proof of Lemma 4.1

First, we develop a spectral bound for *average* reconstruction error of $\mu(\alpha)$, when $\alpha$ is drawn uniformly from the unit interval $[0, 1]$. This is controlled by the tail eigenvalues of the second moment matrix of $\mu(\alpha)$, just as in PCA:

**Lemma C.1.** *Let $\{(\sigma_j, \phi_j)\}_{j=1}^{T}$ be the eigenpairs of $Z$, in decreasing order by eigenvalue. Let $\Psi_k$ be the linear subspace of $\mathbb{R}^T$ spanned by $\{\phi_1, \ldots, \phi_k\}$. Then,*

$$\int_0^1 \|\mu(\alpha) - \mathrm{Proj}_{\Psi_k}(\alpha)\|^2 \, d\alpha \le \sum_{j=k+1}^{T} \sigma_j.$$

*Proof.* Let $r(\alpha)$ denote the residual $\mu(\alpha) - \mathrm{Proj}_{\Psi_k}(\alpha)$, and let $U_r \in \mathbb{R}^{T \times r}$ whose columns are $\phi_1, \ldots, \phi_r$, so that

$$r(\alpha) = \Pi_r \mu(\alpha) := (I - U_r U_r^\top)\mu(\alpha).$$

Write the eigendecomposition $Z_T = U_T \Sigma U_T^\top$. Then,

$$\int_0^1 \|r(\alpha)\|^2 \, d\alpha = \int_0^1 \mathrm{Tr}(r(\alpha) \otimes r(\alpha)) \, d\alpha = \int_0^1 \mathrm{Tr}\left(\Pi_r \mu(\alpha)\mu(\alpha)^\top \Pi_r\right) \, d\alpha$$

$$= \int_0^1 \mathrm{Tr}\left(\Pi_r Z \Pi_r\right) \, d\alpha = \int_0^1 \mathrm{Tr}\left(\Pi_r U_T \Sigma U_T^\top \Pi_r\right) \, d\alpha.$$

Noting that $\Pi_r U_T$ is just $U_T$ with the first $r$ columns set to zero, the integrand becomes $\sum_{j=k+1}^{T} \Sigma_{jj}$, which is the stated bound. $\square$

In fact, this bound *in expectation* turns into a bound for *all* $\alpha$. We show this by noting that $\|r(\alpha)\|^2$ is Lipschitz in $\alpha$, so its maximum over $\alpha \in [0, 1]$ cannot be too much larger than its mean. We state and prove this here:

**Lemma C.2.** *For all $\alpha \in [0, 1]$, it holds that*

$$\|r(\alpha)\|^2 \le \sqrt{6 \sum_{j=k+1}^{T} \sigma_j}.$$

*Proof.* By part (ii) of Lemma F.4, $\|\mu(\alpha)\|^2$ is 3-Lipschitz; since $\Pi_r$ is contractive, $\|r(\alpha)\|^2$ is also 3-Lipschitz. Now, let $R := \max_{0 \le \alpha \le 1} \|r(\alpha)\|^2$. Notice that $R \le \max_{0 \le \alpha \le 1} \|\mu(\alpha)\|^2 \le 1$, by part (i) of Lemma F.4. Subject to achieving a maximum at $R$, the non-negative 3-Lipschitz function $g : [0, 1] \to \mathbb{R}$ with the smallest mean is given by the triangle-shaped function

$$\Delta(\alpha) = \max(R - 3\alpha, 0),$$

for which

$$\int_0^1 \Delta(\alpha)\,d\alpha = R^2/6.$$

In other words,

$$R^2/6 \le \int_0^1 \|r(\alpha)\|^2\,d\alpha.$$

But Lemma C.1 gives a bound on the RHS, so we conclude

$$\max_{\alpha \in [0,1]} \|r(\alpha)\|^2 \le R \le \sqrt{6 \sum_{j=k+1}^{T} \sigma_j},$$

as desired. The stated upper bound on this quantity comes a bound of this spectral tail of the Hankel matrix $Z_T$ (see Lemmas E.2 and E.3); this completes the proof of Lemma 4.1. $\square$

## C.2 Proof of Theorem 3

It remains to apply Lemma 4.1 to the original setting, which will complete the low-rank approximation result of Theorem 3. Indeed, following Section 4.1, we have

$$\zeta_t \stackrel{\text{def}}{=} M_\Theta \tilde{X}_t - \hat{y}_t = \sum_{l=1}^{d}(c_l \otimes b_l) \sum_{i=1}^{T-1} [\tilde{\mu}(\alpha_l) - \mu(\alpha_l)](i) \cdot x_{t-i}.$$

View each of the $n$ coordinates in the inner summation as an inner product between the length-$T$ sequence $\tilde{\mu}(\alpha_l) - \mu(\alpha_l)$ and coordinates $X(j) := (x_1(j), \dots, x_T(j))$, which are entrywise bounded by $R_x$. Then, by Hölder's inequality and Lemma 4.1, we know that this inner product has absolute value at most

$$\|X(j)\|_\infty \|\tilde{\mu}(\alpha_l) - \mu(\alpha_l)\|_1 \le \|X(j)\|_\infty \cdot \sqrt{T} \|\tilde{\mu}(\alpha_l) - \mu(\alpha_l)\|_2 \le O\left(R_x \sqrt{T} \cdot c_1^{-k/\log T} \log^{1/4} T\right),$$

with $c_1 = \sqrt{c_0}$. There are $n$ such coordinates, so this inner summation is a vector with $\ell_2$ norm at most

$$O\left(R_x \sqrt{nT} \cdot c_1^{-k/\log T} \log^{1/4} T\right).$$

Thus, in all, we have

$$\|\zeta_t\|_2 \le O\left(\|B\|_F \|C\|_F R_x \sqrt{nT} \cdot c_1^{-k/\log T} \log^{1/4} T\right).$$

In summary, we have shown that for every system $\Theta$ from which a predictor for the discrete derivative of the LDS arises, there is some $M_\Theta$ whose predictions are pointwise $\|\zeta_t\|_2$-close. This residual bound can be driven down exponentially by increasing the number of filters $k$.

Finally, to get an inequality on the *total* squared error, we compute

$$\sum_{t=1}^{T}\|M_\Theta \tilde{X}_t - y_t\|^2 = \sum_{t=1}^{T}\|\hat{y}_t - y_t + \zeta_t\|^2 \le \sum_{t=1}^{T}\left(\|\hat{y}_t - y_t\|^2 + \|\zeta_t\|^2 + 2\|\hat{y}_t - y_t\|\,\|\zeta_t\|\right)$$

$$\le \sum_{t=1}^{T}\|\hat{y}_t - y_t\|^2 \;+\; O\left((R_\Theta^4 R_x^2 L_y^2 k)\, T^{3/2} n^{1/2} \cdot c_1^{-k/\log T} \log^{1/4} T\right), \quad (2)$$

$$\le \sum_{t=1}^{T}\|\hat{y}_t - y_t\|^2 \;+\; O\left(R_\Theta^4 R_x^2 L_y^2\, T^{5/2} n^{1/2} \cdot c_1^{-k/\log T} \log^{1/4} T\right),$$

where inequality (2) invokes Corollary D.2. Thus, in all, it suffices to choose

$$\frac{k}{\log T} \ge \Omega\left(\log \frac{R_\Theta R_x L_y\, nT}{\varepsilon}\right)$$

to force the $O(\cdot)$ term to be less than $\varepsilon$, noting that the powers of $n$ and $T$ show up as a constant factor in front of the $\log(\cdot)$. This completes the proof. $\square$

# D   Proof of the main regret bound

In this part of the appendix, we follow the proof structure outlined Section 4.2, to establish Theorem 1. The lemmas involved also appear in the proof of the batch variant (Theorem 2).

## D.1   Diameter bound: controlling the comparator matrix

We will show that the $M_\Theta$ that competes with a system $\Theta$ is not too much larger than $\Theta$, justifying the choice of $R_M = \Omega\left(R_\Theta^2 \sqrt{k}\right)$. Of course, this implies that the diameter term in the regret bound is $D_{\max} = 2R_M$. Concretely:

**Lemma D.1.** *For any LDS parameters* $\Theta = (A, B, C, D, h_0 = 0)$ *with* $0 \preccurlyeq A \preccurlyeq I$ *and* $\|B\|_F, \|C\|_F, \|D\|_F, \|h_0\| \leq R_\Theta$, *the corresponding matrix* $M_\Theta \in \hat{\mathcal{H}}$ *(which realizes the relaxation in Theorem 3) satisfies*

$$\|M_\Theta\|_F^2 \leq O\left(R_\Theta^2 \sqrt{k}\right).$$

*Proof.* Recalling our construction $M_\Theta$ in the proof of Theorem 3, we have

- $\|M^{(j)}\|_F \leq \|B\|_F \|C\|_F \cdot \max_{\ell \in [d]} \sigma_j^{-1/4} \langle \phi_j, \mu(\alpha_l) \rangle$, for each $1 \leq j \leq k$.

- $\|M^{(x')}\|_F = \|D\|_F \leq O(R_\Theta)$.

- $\|M^{(x)}\|_F \leq \|B\|_F \|C\|_F + \|D\|_F \leq O(R_\Theta^2)$.

Recall that we do not consider $M^{(y)}$ as part of the online learning algorithm; it is always the identity matrix. Thus, for the purposes of this analysis, it does not factor into regret bounds.

In Lemma E.4, we show that the reconstruction coefficients $\sigma_j^{-1/4} \langle \phi_j, \mu(\alpha_l) \rangle$ are bounded by an absolute constant; thus, those matrices each have Frobenius norm at most $O(R_\Theta^2)$. These terms dominate the Frobenius norm of the entire matrix, concluding the lemma. $\qquad \square$

This has a very useful consequence:

**Corollary D.2.** *The predictions* $\hat{y}_t = M\tilde{X}_t$ *made by choosing* $M$ *such that* $\|M\|_F \leq O(R_\Theta^2 \sqrt{k})$ *satisfy*

$$\|\hat{y}_t - y_t\|^2 \leq O(R_\Theta^4 R_x^2 L_y^2 k).$$

## D.2   Gradient bound and final details

A subtle issue remains: the gradients may be large, as they depend on $\tilde{X}_t$, defined by convolutions of the entire input time series by some filters $\phi_j$. Note that these filters do *not* preserve mass: they are $\ell_2$ unit vectors, which may cause the norm of the part of $\tilde{X}_t$ corresponding to each filter to be as large as $\sqrt{T}$.

Fortunately, this is not the case. Indeed, we have:

**Lemma D.3.** *Let* $\{(\sigma_j, \phi_j)\}_{j=1}^T$ *be the eigenpairs of* $Z$, *in decreasing order by eigenvalue. Then, for each* $1 \leq j, t \leq T$, *it holds that*

$$\|\sigma^{1/4}(\phi_j * X)_t\|_\infty \leq O\left(R_x \log T\right).$$

*Proof.* Each coordinate of $(\sigma^{1/4} \phi_j * X)_t$ is the inner product between $\phi_j$ and a sequence of $T$ real numbers, entrywise bounded by $\sigma_j^{1/4} R_x$. Corollary E.6 shows that this is at most $O(\log T)$, a somewhat delicate result which uses matrix perturbation. $\qquad \square$

Thus, $\tilde{X}_t$ has $nk$ entries with absolute value bounded by $O\left(R_x \log T\right)$, concatenated with $x_t$ and $x_{t-1}$. So, we have:

**Corollary D.4.** *Let $\tilde{X}_t$ be defined as in Algorithm 1, without the $y_{t-1}$ portion. Then,*

$$\|\tilde{X}_t\|_2 \leq O\left(R_x \log T \sqrt{nk}\right).$$

Our bound on the gradient follows:

**Lemma D.5.** *Suppose $\mathcal{M}$ is chosen with diameter $O(R_\Theta^2)$. Then, the gradients satisfy*

$$G_{\max} \stackrel{def}{=} \max_{\substack{M \in \mathcal{M}, \\ 1 \leq t \leq T}} \|\nabla f_t(M)\|_F \leq O\left(R_\Theta^2 R_x^2 L_y \cdot nk^{3/2} \log^2 T\right).$$

*Proof.* We compute the gradient, and apply Lemma D.3:

$$\nabla f_t(M) = \nabla\left(\|y_t - M\tilde{X}_t\|^2\right) = 2(M\tilde{X}_t - y) \otimes \tilde{X}_t,$$

so that

$$
\begin{aligned}
\|\nabla f_t(M)\|_F &= 2\|M\tilde{X}_t - y_t\|_2 \cdot \|\tilde{X}_t\|_2 \\
&\leq 2\left(\|M\|_F \|\tilde{X}_t\|_2 + L_y\right)\|\tilde{X}_t\|_2 \\
&\leq 2\left(\left(R_\Theta^2 \sqrt{k}\right)\left(R_x \log T \sqrt{nk}\right) + L_y\right)\left(R_x \log T \sqrt{nk}\right) \\
&\leq O\left(R_\Theta^2 R_x^2 L_y \cdot nk^{3/2} \log^2 T\right),
\end{aligned}
$$

as desired. $\square$

### D.3 Assembling the regret bound

Using Lemma 4.2, collecting all terms from Lemmas D.1 and D.5, we have in summary

$$
\begin{aligned}
D_{\max} G_{\max} &= O\left(R_\Theta^2 \sqrt{k}\right) \cdot O\left(R_\Theta^2 R_x^2 L_y \cdot nk^{3/2} \log^2 T\right) \\
&= O\left(R_\Theta^4 R_x^2 L_y n k^2 \log^2 T\right).
\end{aligned}
$$

To compete with systems with parameters bounded by $R_\Theta$, in light of Theorem 3, $k$ should be chosen to be $\Theta\left(\log^2 T \log(R_x L_y R_\Theta n)\right)$. It suffices to set the relaxation approximation error $\varepsilon$ to be a constant; in the online case, this is not the bottleneck of the regret bound. In all, the regret bound from online gradient descent is

$$\mathrm{Regret}(T) \leq O\left(R_\Theta^4\, R_x^2\, L_y\, \log^2(R_\Theta R_x L_y n) \cdot n\sqrt{T} \log^6 T\right),$$

as claimed. $\square$

### D.4 Diminishing effect of the initial hidden state

Finally, we show that $h_0$ is not significant in this online setting, thereby proving a slightly more general result. Throughout the rest of the analysis, we considered the comparator $\Theta^*$, which forces the initial hidden state to be the zero vector. We will show that this does not make much worse predictions than $\Theta^{**}$, which is allowed to set $\|h_0\|_2 \leq R_\Theta$. We quantify this below:

**Lemma D.6.** *Relaxing the condition $h_0 = 0$ for the comparator in Theorem 1 increases the regret (additively) by at most*

$$O\left(R_\Theta^4 R_x L_y \log(R_\Theta R_x L_y n) \log^2 T\right).$$

*Proof.* First, an intuitive sketch: Lemma F.1 states that for any $\alpha$, there is an "envelope" bound $\mu(\alpha)(t) \leq \frac{1}{t+1}$. This means that the influence of $h_0$ on the derivative of the impulse response function decays as $1/t$; thus, we can expect the total "loss of expressiveness" caused by forcing $h_0 = 0$ to be only logarithmic in $T$.

Indeed, with a nonzero initial hidden state, we have

$$\hat{y}_t - y_{t-1} = (CB + D)x_t - Dx_{t-1} + \sum_{i=1}^{T-1} C(A^i - A^{i-1})Bx_{t-i} + C(A^t - A^{t-1})h_0.$$

Let $\hat{y}_1, \ldots, \hat{y}_T$ denote the predictions made by an LDS $\Theta^{**} = (A, B, C, D, h_0)$ whose; $\hat{y}_1^{\emptyset}, \ldots, \hat{y}_T^{\emptyset}$ denote the predictions made by the LDS with the same $(A, B, C, D)$, but $h_0$ set to 0. Then, we have

$$\|\hat{y}_t - \hat{y}_t^{\emptyset}\| = \|C(A^t - A^{t-1})h_0\| = \left\| \sum_{l=1}^{d} C \left[ \mu(\alpha_l)(t) \cdot e_l \otimes e_l \right] h_0 \right\|$$

$$\leq \frac{\|C\|_F \|h_0\| \sqrt{n}}{t} \leq \frac{R_{\Theta}^2 \sqrt{n}}{t}.$$

Thus we have, for vectors $u_t$ satisfying $\|u_t\| \leq R_{\Theta}^2 / t$:

$$\sum_{t=1}^{T} \|\hat{y}_t^{\emptyset} - y_t\|^2 = \sum_{t=1}^{T} \|\hat{y}_t + u_t - y_t\|^2 \leq \sum_{t=1}^{T} \|\hat{y}_t - y_t\|^2 + \|u_t\|^2 + 2\|\hat{y}_t - y_t\| \|u_t\|$$

$$\leq \sum_{t=1}^{T} \|\hat{y}_t - y_t\|^2 + O\left(R_{\Theta}^4 n\right) + O\left((R_{\Theta}^2 R_x L_y \sqrt{k}) \cdot R_{\Theta}^2 \sqrt{n} \log T\right)$$

$$\leq \sum_{t=1}^{T} \|\hat{y}_t - y_t\|^2 + O\left(R_{\Theta}^4 R_x L_y \log(R_{\Theta} R_x L_y n) \, n \log^2 T\right),$$

where the inequalities respectively come from Cauchy-Schwarz, Lemma F.1, and Lemma D.2. This completes the proof. $\qquad\square$

Thus, strengthening the comparator by allowing a nonzero $h_0$ does not improve the asymptotic regret bound from Theorem 1.

# E  Properties of the Hankel matrix $Z_T$

In this section, we show some technical lemmas about the family of Hankel matrices $Z_T$, whose entries are given by

$$Z_{ij} = \frac{2}{(i+j)^3 - (i+j)}.$$

## E.1  Spectral tail bounds

We use the following low-approximate rank property of positive semidefinite Hankel matrices, from [BT16]:

**Lemma E.1** (Cor. 5.4 in [BT16])**.** *Let $H_n$ be a psd Hankel matrix of dimension $n$. Then,*

$$\sigma_{j+2k}(H_n) \leq 16 \left[ \exp\left( \frac{\pi^2}{4\log(8\lfloor n/2 \rfloor / \pi)} \right) \right]^{-2k+2} \sigma_j(H_n).$$

Note that the Hankel matrix $Z_T$ is indeed positive semidefinite, because we constructed it as

$$Z = \int_0^1 \mu(\alpha) \otimes \mu(\alpha) \, d\alpha,$$

for a certain $\mu(\alpha) \in \mathbb{R}^T$. Also, note that at no point do we rely upon $Z_T$ being positive definite or having all distinct eigenvalues, although both seem to be true.

The first result we need is an exponential decay of the tail spectrum of $Z$.

**Lemma E.2.** *Let $\sigma_j$ be the $j$-th top singular value of $Z := Z_T$. Then, for all $T \geq 10$, we have*

$$\sigma_j \leq \min\left(\frac{3}{4}, K \cdot c^{-j/\log T}\right),$$

*where $c = e^{\pi^2/4} \approx 11.79$, and $K < 10^6$ is an absolute constant.*

*Proof.* We begin by noting that for all $j$, $\sigma_j \leq \text{Tr}(Z) = \sum_{i=1}^{T} \frac{1}{(2i)^3 - 2i} < \sum_{i=1}^{\infty} \frac{1}{4i^3} < \frac{3}{4}$.

Now, since $T \geq 10$ implies $8\lfloor T/2 \rfloor / \pi > T$, we have

$$\sigma_{2+2k} \leq \sigma_{1+2k} < 12 \cdot \left[\exp\left(\frac{\pi^2}{2\log T}\right)\right]^{-k+1}$$
$$< 1680 \cdot c^{-2k/\log T}.$$

Thus, we have that for all $j$,

$$\sigma_j < 1680 \cdot c^{-(j-2)/\log T} < 235200 \cdot c^{-j/\log T},$$

completing the proof. $\qquad\square$

We also need a slightly stronger claim: that all spectral gaps are large. Lemma E.2 does not preclude that there are closely clustered eigenvalues under the exponential tail bound. In fact, this cannot be the case:

**Lemma E.3.** *Let $\sigma_j$ be the $j$-th top singular value of $Z := Z_T$. Then, if $T \geq 60$, we have*

$$\sum_{j'>j} \sigma_{j'} < 400\log T \cdot \sigma_j.$$

*Proof.* For convenience, define $\sigma_j := 0$ when $j \geq T$. Picking $k = 4$ and using Lemma E.1, we have that

$$\beta_j := \sum_{q=1}^{T} \sigma_{j+4q} < 16\sigma_j \sum_{q=1}^{\infty} \left[\exp\left(\frac{-\pi^4}{4\log T}\right)\right]^q = 16\sigma_j \cdot \frac{1}{1 - \exp\left(\frac{-\pi^4}{4\log T}\right)}$$
$$< 100\log T \cdot \sigma_j,$$

where the last inequality follows from the fact that

$$\frac{1}{1 - e^{-x}} < \frac{6}{x}$$

whenever $x < 6$, and setting $x := \frac{-\pi^4}{4\log T} \leq \frac{-\pi^4}{4\log 60} < 6$.

Thus, we have

$$\sum_{j'>j} \sigma_{j'} = \beta_j + \beta_{j+1} + \beta_{j+2} + \beta_{j+3} < 4\beta_j < 400\log T \cdot \sigma_j,$$

as desired. $\qquad\square$

## E.2 Decaying reconstruction coefficients

To show a bound on the entries of $M_\Theta$, we need the following property of $Z_T$:

**Lemma E.4.** *For any $0 \leq \alpha \leq 1$ and $1 \leq j \leq T$, we have*

$$|\langle \phi_j, \mu(\alpha)\rangle| \leq 6^{1/4}\, \sigma_j^{1/4}.$$

*Proof.* We have

$$\int_0^1 \langle \phi_j, \mu(\alpha) \rangle^2 \, d\alpha = \int_0^1 \phi_j^T \left( \mu(\alpha) \otimes \mu(\alpha) \right) \phi_j$$
$$= \phi_j^T Z_T \phi_j = \sigma_j.$$

Thus, we have a bound on the expectation of the squared coefficient, when $\alpha$ is drawn uniformly from $[0, 1]$. We proceed with the same argument as was used to prove Lemma C.2: since $\|\mu(\alpha)\|^2$ is 3-Lipschitz in $\alpha$, so is $\langle \phi_j, \mu(\alpha) \rangle^2$ (since projection onto the one-dimensional subspace spanned by $\phi_j$ is contractive). Thus it holds that

$$\max_{\alpha \in [0,1]} \langle \phi_j, \mu(\alpha) \rangle^2 \le \sqrt{6\sigma_j},$$

from which the claim follows. $\qquad\qquad\qquad\qquad\qquad\qquad\qquad\qquad\qquad\qquad\qquad\qquad\qquad$ $\square$

### E.3 Controlling the $\ell_1$ norms of filters

To bound the size of the convolutions, we need to control the $\ell_1$ norm of the eigenvectors $\phi_j$ with a tighter bound than $\sqrt{T}$. Actually, we prove a more general result, bounding the $\ell_2 \to \ell_1$ subordinate norm of $Z^{1/4}$:

**Lemma E.5.** *Let $Z := Z_T$. Then, for every $T > 0$, and $v \in \mathbb{R}^n$ with $\|v\|_2 = 1$, we have*

$$\|Z^{1/4}v\|_1 \le 2 + 2\log_2 T.$$

*Proof.* We take the following steps:

  (i) Start with a constant $T_0$; the subordinate norm of $Z_{T_0}$ is clearly bounded by a constant.

 (ii) Argue that doubling the size of the matrix $(T \mapsto 2T)$ comprises only a small perturbation, which will only affect the eigenvalues of the matrix by a small amount. This will show up in the subordinate norm as an additive constant.

(iii) Iterate the doubling argument $O(\log T)$ times to reach $Z_T$ from $Z_{T_0}$, to conclude the lemma.

The only nontrivial step is (ii), which we prove first. Consider the doubling step from $T$ to $2T$. Let $Z$ denote the $2T$-by-$2T$ matrix which has $Z_T$ as its upper left $T$-by-$T$ submatrix, and zero everywhere else. Let $Z'$ denote $Z_{2T}$, and call $E = Z' - Z$, which we interpret as the matrix perturbation associated with doubling the size of the Hankel matrix.

Notice that when $T \ge 2$, $E$ is entrywise bounded by $\frac{2}{(T+2)^3 - (T+2)} \le \frac{2}{T^3}$, which we call $e_{\max}$ for short. Then, $\|E\|_{\text{op}}$ is at most $Te_{\max} \le \frac{2}{T^2}$.

Hence, by the generalized Mirsky inequality of [Aud14] (setting $f(x) = x^{1/4}$), we have a bound on how much $E$ perturbs the fourth root of $Z$:

$$\|Z^{1/4} - Z'^{1/4}\|_2 \le \|E\|_2^{1/4} \le \left( \frac{2}{T^2} \right)^{1/4} < \frac{2}{\sqrt{T}}.$$

Thus we have

$$\|Z'^{1/4}\|_{2\to1} \le \|Z^{1/4}\|_{2\to1} + \|Z^{1/4} - Z'^{1/4}\|_{2\to1}$$
$$\le \|Z^{1/4}\|_{2\to1} + \sqrt{T} \cdot \|Z^{1/4} - Z'^{1/4}\|_2$$
$$\le \|Z^{1/4}\|_{2\to1} + \sqrt{T} \cdot \frac{2}{\sqrt{T}}$$
$$= \|Z^{1/4}\|_{2\to1} + 2.$$

Thus, doubling the dimension increases the subordinate norm by at most a constant. We finish the argument: start at $T_0 = 2$, for which it clearly holds that

$$\|Z_2^{1/4}\|_{2\to 1} < \sqrt{2}\|Z_2^{1/4}\|_F < \sqrt{2}\|Z_4\|_F < 2.$$

Noting that the norm is clearly monotonic in $T$, we repeat the doubling argument $\lfloor \log_2 T \rfloor$ times, so that

$$\|Z_T^{1/4}\|_{2\to 1} \leq \|Z_{2\cdot 2^{\lfloor \log_2 T \rfloor}}^{1/4}\|_{2\to 1} < \|Z_2^{1/4}\|_{2\to 1} + 2\lfloor \log_2 T \rfloor < 2 + 2\log_2 T,$$

as claimed. $\qquad\square$

We give an alternate form here:

**Corollary E.6.** *Let $(\sigma_j, \phi_j)$ be the $j$-th largest eigenvalue-eigenvector pair of $Z$. Then,*

$$\|\phi_j\|_1 \leq O\left(\frac{\log T}{\sigma_j^{1/4}}\right).$$

# F   Properties of $\mu(\alpha)$

Throughout this section, fix some $T \geq 1$; then, recall that $\mu(\alpha) \in \mathbb{R}^T$ is defined as the vector whose $i$-th entry is $(1-\alpha)\alpha^{i-1}$. At various points, we will require some elementary properties of $\mu(\alpha)$, which we verify here.

**Lemma F.1** (1/t envelope of $\mu$). *For any $t \geq 0$ and $0 \leq \alpha \leq 1$, it holds that*

$$(1-\alpha)\alpha^t \leq \frac{1}{t+1}.$$

*Proof.* Setting the derivative to zero, the global maximum occurs at $\alpha^* = \frac{t}{t+1}$. Thus,

$$(1-\alpha^*)(\alpha^*)^t = \frac{1}{t+1}\left(1 - \frac{1}{t+1}\right)^t \leq \frac{1}{t+1},$$

as claimed. $\qquad\square$

**Corollary F.2.** *Let $T \geq 1$. For $t = 1, \ldots, T$, let $\alpha_t \in [0,1]$ be different in general. Then,*

$$\sum_{t=1}^{T}(1-\alpha_t)\alpha_t^{t-1} \leq H_n = O(\log T),$$

*where $H_n$ denotes the $n$-th harmonic number.*

**Lemma F.3** ($\ell_1$-norm is small). *For all $T \geq 1$ and $0 \leq \alpha \leq 1$, we have*

$$\|\mu(\alpha)\|_1 \leq 1.$$

*Proof.* We have

$$\|\mu(\alpha)\|_1 = (1-\alpha)\sum_{t=1}^{T}\alpha^{t-1} \leq (1-\alpha)\sum_{t=1}^{\infty}\alpha^{t-1} = 1,$$

proving the claim. $\qquad\square$

**Lemma F.4** ($\ell_2$-norm is small and Lipschitz). *For all $T \geq 1$ and $0 \leq \alpha \leq 1$, we have*

   *(i) $\|\mu(\alpha)\|^2 \leq 1$.*

   *(ii) $\left|\frac{d}{d\alpha}\|\mu(\alpha)\|^2\right| \leq 3$.*

*Proof.* For the first inequality, compute

$$\|\mu(\alpha)\|^2 = \sum_{i=1}^{T} \left((\alpha - 1)\alpha^{i-1}\right)^2 = \sum_{i=1}^{T} \alpha^{2i} - 2\alpha^{2i-1} + \alpha^{2i-2}$$

$$= \frac{(\alpha^2 - 2\alpha + 1)(1 - \alpha^{2T})}{1 - \alpha^2} = \frac{(1 - \alpha)(1 - \alpha^{2T})}{1 + \alpha} \le 1.$$

For the second, differentiate the closed form to obtain

$$\left|\frac{d}{d\alpha}\|\mu(\alpha)\|^2\right| = \left|\frac{2(\alpha^T - 1) + T\alpha^{T-1}(\alpha^2 - 1)}{(1 + \alpha)^2}\right| \le \frac{2(1 - \alpha^T) + T\alpha^{T-1}(1 - \alpha^2)}{(1 + \alpha)^2}$$

$$= \frac{2 - \alpha^T}{(1 + \alpha)^2} + \frac{T\alpha^{T-1}(1 - \alpha)}{1 + \alpha} \le 2 + T\alpha^{T-1}(1 - \alpha) \le 3,$$

where the final inequality uses Lemma F.1. $\square$

### F.1 The Lipschitzness of a true LDS

We claim in Section 2.2 that $L_y$, the Lipschitz constant of a true LDS, is bounded by $\|B\|_F\|C\|_F R_x$. We now prove this fact, which is a consequence of the above facts.

**Lemma F.5.** *Let $\Theta = (A, B, C, D, h_0)$ be a true LDS, which produces outputs $y_1, \ldots, y_T$ from inputs $x_1, \ldots, x_T$ by the definition in the recurrence, without noise. Let $0 \preccurlyeq A \preccurlyeq I$, and $\|B\|_F, \|C\|_F, \|D\|_F, \|h_0\| \le R_\Theta$. Then, we have that for all $t$,*

$$\|y_t - y_{t-1}\| \le O(R_\Theta^2 R_x).$$

*Proof.* We have that for all $1 \le t \le T$,

$$\|y_t - y_{t-1}\| = \left\|(CB + D)x_t - Dx_{t-1} + \sum_{i=1}^{T-1} C(A^i - A^{i-1})Bx_{t-i} + C(A^t - A^{t-1})h_0\right\|$$

$$\le (\|B\|_F\|C\|_F + 2\|D\|_F)R_x + \|B\|_F\|C\|_F R_x + \frac{\|C\|_F\|h_0\|}{t},$$

where the inequality on the second term arises from Lemma F.3 and the inequality on the third from Lemma F.2. This implies the lemma. $\square$

## Footnotes

[1]The distinction between measuring total vs. mean squared error is hidden in the constant in front of the $\log T$ when choosing the number of filters $k$.