[Reviews · NeurIPS 2017]

Reviewer 1



The paper addresses an interesting issue and the proposed approach has merit. However, the analysis is based on a number of strong conditions that significantly limit the class of LDS that are considered. One of the assumptions is that the A matrix is symmetric and strictly psd for which the authors provide some justification. A stronger assumption is that the input is bounded entrywise, which is much harder to justify. Finally, stability of an LDS is usually defined through the spectral radius norm. On the other hand, in this paper the authors work with the operator norm that is significantly more restrictive. There is no discussion of the 2nd and 3rd assumption and which LDS systems fail to cover.

Reviewer 2



Linear dynamical systems are a mainstay of control theory. Unlike the empirical morass of work that underlies much of machine learning work, e.g. deep learning, where there is little theory, and an attempt to produce general solutions of unknown reliability and quality, control theorists wisely have chosen an alternative course of action, where they focused on a simple but highly effective linear model of dynamics, which can be analyzed extremely deeply. This led to the breakthrough work many decades ago of Kalman filters, without which the moon landing would have been impossible. This paper explores the problem of online learning (in the regret model) of dynamical systems, and improves upon previous work in this setting that was restricted to the single input single output (SISO) case [HMR 16]. Unlike that paper, the present work shows that regret bounded learning of an LDS is possible without making assumptions on the spectral structure (polynomially bounded eigengap), and signal source limitations. The key new idea is a convex relation of the original non-convex problem, which as the paper shows, is "the central driver" of their approach. The basic algorithm is a variant of the original projected gradient method of Zinkevich from 2003, The method is viewed as a online wave filtered regression method (Algorithm 1), where pieces of the dynamical system are learned over time and stitched together into an overall model. The paper shows that optimization over linear maps is appropriate convex relaxation of the original objective function,. The paper is well written and has an extensive theoretical analysis. It represents a solid step forward in the learning of dynamical systems models, and should spur further work on more difficult cases. The convex relaxation trick might be useful in other settings as well. Some questions: 1. Predictive state representations (PSRs) are widely used in reinforcement learning to model learning from partially observed states. Does your work have any bearing on the learnability of PSRs, a longstanding challenge in the field. 2. There is quite a bit of work on kernelizing Kalman filters in various ways to obtain nonlinear dynamical systems. Does your approach extend to any of these extended models? The approach is largely based on the simple projected gradient descent approach, but one wonders whether proximal gradient tricks that are so successful elsewhere (e.g, ADAGRAD, mirror prox etc.) would also be helpful here. In other words, can one exploit the geometry of the space to accelerate convergence?

Reviewer 3



An interesting and very strong paper. The results and proofs are extremely relevant to the NIPS audience. The paper is clearly written and provides a method for improving bounds in online learning of LDS in an efficient manner, as stated. The example applications show significant practical improvement over EM. Suggestions: - Comparisons to other sys-id methods would be welcome (e.g. least squares, GLS, etc.) - More information in the main paper on the nature of adversarial noise that can be tolerated would be helpful. An example would be ideal. This is presumably linked to performance on the nonlinear system considered. - LIne 207 is missing words.